# Quantification of Non-Motor Symptoms in Parkinsonian Cynomolgus Monkeys

**DOI:** 10.3390/brainsci13081153

**Published:** 2023-08-01

**Authors:** Yu Bao, Chaoning Gan, Zuyue Chen, Zhongquan Qi, Zhiqiang Meng, Feng Yue

**Affiliations:** 1Shenzhen Key Laboratory of Drug Addiction, Shenzhen Neher Neural Plasticity Laboratory, The Brain Cognition and Brain Disease Institute, Shenzhen Institute of Advanced Technology, Chinese Academy of Sciences, Shenzhen 518055, China; yubao0558@126.com (Y.B.);; 2Department of Neurobiology, Beijing Institute of Geriatrics, Xuanwu Hospital of Capital Medical University, Beijing 100053, China; 3University of Chinese Academy of Sciences, Beijing 100049, China; 4Medical College of Guangxi University, Nanning 530003, China; 5CAS Key Laboratory of Brain Connectome and Manipulation, Shenzhen Institute of Advanced Technology, Chinese Academy of Sciences, Shenzhen 518055, China; 6Shenzhen-Hong Kong Institute of Brain Science-Shenzhen Fundamental Research Institutions, Shenzhen 518055, China; 7Key Laboratory of Biomedical Engineering of Hainan Province, School of Biomedical Engineering, Hainan University, Haikou 570228, China

**Keywords:** Parkinson’s, monkey, non-motor symptoms, anxiety, sleep, cognition

## Abstract

Background: Parkinson’s disease (PD) is a neurodegenerative disorder that features motor and non-motor deficits. The use of 1-methyl-4-phenyl-1,2,3,6-tetrahydropyridine (MPTP)-induced dopamine neuron degeneration has been widely practiced to produce reliable animal models of PD. However, most previous preclinical studies focused on motor dysfunction, and few non-motor symptoms were evaluated. Thus far, there is a lack of comprehensive investigations of the non-motor symptoms in animal models. Objectives: In this study, we aim to use a battery of behavioral methods to evaluate non-motor symptoms in MPTP-induced non-human primate PD models. Methods: Cognitive function, sleep, and psychiatric behaviors were evaluated in MPTP-treated cynomolgus monkeys. The tests consisted of a delayed matching-to-sample (DMTS) task, the use of a physical activity monitor (PAM), an apathy feeding task (AFT), the human intruder test (HIT), novel fruit test (NFT), and predator confrontation test (PCT). In addition, we tested whether the dopamine receptor agonist pramipexole (PPX) can improve these non-motor symptoms. Results: The present results show that the MPTP-treated monkeys exhibited cognitive deficits, abnormal sleep, and anxiety-like behaviors when compared to the control monkeys. These symptoms were relieved partially by PPX. Conclusions: These results suggest that MPTP-induced PD monkeys displayed non-motor symptoms that were similar to those found in PD patients. PPX treatment showed moderate therapeutic effects on these non-motor symptoms. This battery of behavioral tests may provide a valuable model for future preclinical research.

## 1. Background

Parkinson’s disease (PD) is a progressive neurodegenerative disease that affects about 3% of the population aged 65 years or older [1]. Its pathological hallmark consists of the α-synuclein-containing Lewy bodies and Lewy neurites with cell loss in the sub- stantia nigra and other brain areas [2]. The diagnosis of PD is currently dependent on the presence of motor deficits including bradykinesia, rigidity, and a resting tremor, unilaterally or asymmetrically [3,4]. In fact, non-motor symptoms (NMSs) such as cognitive deficits, sleep disorders, anxiety, etc., co-occur or even precede the onset of motor symptoms [4]. Remarkably, NMSs are also key factors that contribute to a low quality of life and the progression of overall disability [5,6,7].

In recent decades, a neurotoxin 1-methyl-4-phenyl-1,2,3,6-tetrahydropyridine (MPTP)-induced non-human primate (NHP) PD model has been recognized as a reliable preclinical animal model [8]. MPTP-lesioned monkeys exhibit neuroanatomical and behavioral symptoms resembling those seen in PD patients, particularly the motor deficits [9]. However, non-motor symptoms in MPTP-induced PD monkeys have not been fully investigated [4]. Most studies only involved one or two specific non-motor symptoms [10]. Compared to rodents, NHPs are more suitable for comprehensive assessments of non-motor symptoms [11]. In this study, we investigated a battery of non-motor symptoms in MPTP-induced PD cynomolgus monkeys. The non-motor assessment included cognitive function, sleep, and psychiatric behaviors. One of the reasons that non-motor behaviors were not well studied is that these behaviors often rely on normal motor function. The present data were collected eight years after MPTP administration; thus, the monkeys had eight years of adaption to motor dysfunction.

Pramipexole (PPX) is FDA-approved non-ergot D2/D3 dopamine receptor agonist [12]. It has been shown to improve motor symptoms, psychiatric symptoms, unified Parkinson’s disease rating scale (UPDRS) scores [13,14,15,16], and depressive symptoms [17] in PD patients. The neuroprotective effects have been reported in several studies [18,19]. In the current study, we tested the effects of PPX on non-motor symptoms in NHP PD models.

Our objective was to investigate various behavioral methods for objectively measuring non-motor symptoms in MPTP-induced monkeys. Additionally, we aimed to assess the therapeutic effects of pramipexole on non-motor symptoms of PD.

## 2. Materials and Methods

### 2.1. Animals

Ten male cynomolgus monkeys (age: 15 ± 1.5 years; weight: 8.5 ± 1 kg) were used in this study. All experiments were conducted at Wincon TheraCells Biotechnologies Co., Ltd., Nanning, China. A group of five monkeys were injected with MPTP unilaterally through the left internal carotid eight years ago. Another group of five age-matched naive monkeys were injected with the vehicle. All animals were individually housed in standard laboratory conditions (a room temperature of 23~27 °C, humidity of 40~75%) in a 12 h light/12 h dark cycle (lights on from 7 AM to 7 PM) with ad libitum access to water. The primate diet (China standard (GB) 14924. 8-2001) was provided twice daily, and fresh fruit/vegetables were given once daily at noon. The health condition was monitored daily, and the environment was enriched with various rubber toys. All efforts were made to limit the animals’ stress. All experimental protocols were approved by the Institutional Animal Care and Use Committee (AAALAC: WD-0272009).

### 2.2. Drugs

Pramipexol hydrochloride (Booehringer Ingelheim Pharma GmbH & Co KG, Biberach, Germany) was purchased from Nanning Hospital (Nanning, China). The doses used in the current study were 0.375 mg/day for 3 days, 0.75 mg/day for 3 days, 2.25 mg/day for 5 days, and 3.375 mg/day for 7 days, which were gradually increased as per the instructions [20]. Eight years after the induction of MPTP, the treatment protocol was modified from the instructions of using this drug in human patients.

### 2.3. Behavioral Tests

The behavioral tests in the current study included a delayed matching-to-sample (DMTS) task, the use of a physical activity monitor (PAM), an apathy feeding task (AFT), the human intruder test (HIT), novel fruit test (NFT), and predator confrontation test (PCT).

#### 2.3.1. Delayed Matching-To-Sample Task

The delayed matching-to-sample (DMTS) task was modified from the Wisconsin General Test Apparatus (WGTA), which is often used to test learning and memory in monkeys [21]. The DMTS was performed in a quiet testing room. 

The device features three food wells with covers that are used for hiding food rewards and providing clues (Figure 1a). The monkey was transferred into a testing cage equipped with an opaque sliding board. The monkeys were well-trained to perform the DMTS task (the correct rate was ≥80%) before the tests. Each trial comprised three phases: a cue presentation phase, a delay phase, and a responding phase. During the cue presentation phase, a visual cue was presented on the middle cover, and the opaque board was lifted for ~5 s to make sure that the animal saw the cue. The delay phase started from the drop of the opaque board and lasted for 5, 10, 15, or 30 s randomly. During the delay, a food reward (a small piece of fruit or nut) was placed in the left or right well (randomly selected). The previously presented cue was put on top of the food well cover. A distractive cue was placed on top of the other cover. After the delay, the opaque board was lifted. The monkey was able to open the cover and take the food reward. There were 80 trials in the DMTS test (20 trials for each delay). The correct responses were recorded, and the percentage of correct responses was used to evaluate working memory.

#### 2.3.2. Physical Activity Monitor

A physical activity monitor (PAM) measures locomotor activity, which indicates the sleep status indirectly during night. The PAMs used to record physical activity were omnidirectional accelerometers (Actical Activity Monitor; MiniMitter Inc., Bend, OR, USA). The accelerometer was placed in a small stainless-steel box, and the box was attached to a loose-fitting collar on the neck (Figure 2a). The accelerometer was deployed one week before the test. The data in the accelerometer were downloaded using ActiReader (MiniMitter Inc., Bend, OR, USA), and three consecutive days of data were analyzed. The locomotor activity count was determined by the total number of moving counts per minute. Nocturnal activity between 7:00 PM and 7:00 AM was analyzed in this study.

#### 2.3.3. Apathy Feeding Task

The apathy feeding task (AFT) measures the degree and frequency of coaxing required to motivate an animal to attempt to feed [22]. The animal was restrained in a primate chair. A piece of food was offered by a research assistant the animal was familiar with (Figure 3a). The food was alternatively placed on two sides. If the monkey failed to reach for the food in 5 s, the food was discarded and substituted with another piece of food. If the monkey failed to reach for the fruit again, the fruit was placed in the monkey’s hand. One successful trial was defined as fetching the food from experimenter’s hand. The tests were videotaped, and the behavior was scored offline later. The duration of observation was the time from the presentation of the food to the onset of reaching out for the food, and the duration of execution was the time from reaching out for the food to the time of placing the food into the mouth. 

#### 2.3.4. Behavioral Tests for Anxiety

We used the human intruder test (HIT), novel fruit test (NFT), and predator confrontation test (PCT) to measure the anxiety levels of the animals.

##### Human Intruder Test

The HIT was modified from Kalin and colleagues’ study to assess behavioral responsiveness to a potential threatening or a nonthreatening social stimulus in rhesus monkeys [23]. The test consisted of four phases: baseline, profile, stare, and back, with each phase lasting 2 minutes (Figure 4a). A camera was used to record the behavioral responses. Following the baseline phase, a human intruder wearing a mask and a cloak entered the testing room. The human intruder obliquely presented their profile to the monkey (profile), then turned to face the monkey (stare). During the back phase, the intruder turned their back to the monkey. The intruder left the testing room after the tests.

The video was processed using Squared5 (MPEG Streamclip software) and compressed to 30 frames/s for behavioral scoring. The HIT-induced anxious behavior depended on the orientation of the human intruder and the specific threat level posed by each orientation [24]. Increased vigilance, excessive fear, and other context-dependent anxiety-like behaviors were scored to evaluate the anxiety level [25]. The scored behaviors were movement (the whole body moves from one position to another), sitting up (placing the buttocks on the cage and supporting the body weight), tactile/oral exploration (tactile or oral manipulation of the cage with a finger/mouth), self-grooming (picking up dirt or brushing hair with hands or mouth), lip-smacking (pursing and moving the lips together to produce a smacking sound) [26], backing into the cage (subject positioning itself with at least three limbs in the back half of the cage), freezing (remaining immobile for longer than 2 s) [23,26], scratching (moving digits quickly through fur), growling (grunting, short, understated vocalizations), cage shaking (grabbing and shaking the cage with hands or feet), mouth-opening (opening the mouth in an “O” shape stare), screaming (a loud, high-pitched sound), and the total number of times a fear grimace was presented (a large, grin-like facial expression showing the teeth) [26], grinding the teeth (upper and lower teeth moving noisily together), and yawning (opening mouth and showing clenched teeth) [27]. The scored behaviors are shown in Table 1.

##### Novel Fruit Test

The NFT was performed after the HIT to assess each monkey’s motivation to explore novel objects. The assumption of the test is that animals with lower levels of anxiety will exhibit more explorative behaviors. The paradigm used in this study was modified from Williamson et al. (Figure 4b) [28]. During each trial, a piece of novel fruit (which the animals had never had before) was placed in the cage for 2 min. Afterwards, a piece of a familiar fruit (a fruit used as food enrichment) was placed in the cage for 2 min. There were five trials in the test. The monkeys’ behavior was videotaped. The consumption rate, observation time, and execution time were calculated offline.

##### Predator Confrontation Test

The PCT was performed right after the NFT test. Barros and colleagues have shown that stimulus with specific features of natural predators induces fear and anxiety-like reactions in non-human primates [29]. A potential predator model (i.e., a snake model) and a familiar object (i.e., fruit) were placed in the front part of the cage, one on the left side and the other on the right side (Figure 4c). There were five testing trials for each monkey. The responses of each animal were videotaped and scored. Withdrawal behavior (retreating to the back of the cage) and the rate of consumption of fruit were used to determine the fear and anxiety levels.

### 2.4. Statistical Analysis

All data were analyzed using SPSS 22. 0. The data of the AFT, PAM, NFT and PCT were log-transformed and passed a normal distribution test (Shapiro–Wilk test, *p* > 0.05); a multi-way ANOVA was used to analyze the duration of the observation data from the AFT. A one-way ANOVA was used to analyze the PAM data, and the data are expressed as means ± SDs. Nonparametric tests were used to analyze the DMTS task data, the duration of execution data from the AFT, and the HIT data (Shapiro–Wilk test, *p* < 0.05). Differences between DMTS, HIT, NFT, and PCT and the duration of execution in the AFT were assessed using the Kruskal–Wallis H Test, and the data were expressed as medians (interquartile interval). A *p*-value of less than 0.05 was used to determine statistical significance.

## 3. Results

### 3.1. PD Monkeys Showed Working Memory Deficits in DMTS Task

Four out of five healthy monkeys were successfully trained to complete the DMTS task; the correct ratios were above 80% without delay. The PD monkeys had noticeable unilateral limb disabilities. Four out of five PD monkeys were successfully trained before the tests. Two of them completed the task using only the normal limbs. The correct ratio for the control monkeys was significantly higher than for PD monkeys for all delays (5 s delay: 0.90(0.11) vs. 0.53, H = −2.160, *p* = 0.031; 10 s delay: 0.97(0.17) vs. 0.60, H = −2.160, *p* = 0.031; 15 s delay: 0.87(0) vs. 0.40, H (0.000) = −2.366, *p* = 0.018; 30 s delay: 0.90(0.06) vs. 0.47, H = −2.181, *p* = 0.029). After the PPX treatment, the correct ratio was modestly improved (5 s: 0.53 vs. 0.75(0.36), H = 0.825, *p* = 0.031; 10 s: 0.60 vs. 0.70(0.32), H = 1.101, *p* = 0.487; 15 s: 0.40 vs. 0.50(0.37), H = 0.741, *p* = 0.261; 30 s: 0.47 vs. 0.65(0.34), H = 1.080, *p* = 0.268). These data indicate that working memory in PD monkeys was impaired, and PPX treatment did not improve the performance significantly (Figure 1b).

### 3.2. PD Monkeys Showed More Nocturnal Activity

Each monkey’s activity was continuously monitored in its home cage. The total nocturnal activity in PD monkeys (2752.40 ± 1906.57) was higher than in control monkeys (2000.55 ± 1363.78), but not to a significant level (F (2,53) = 1.289, *p* = 0.114, Figure 2c). From 21:00 to 23:00, the hourly activity in the PD monkeys was significantly higher (21:00–22:00: 92.10 ± 126.42 vs. 257.30 ± 284.26, F (2,53) = 3.482, *p* = 0.044; 22:00–23:00: 98.05 ± 138.42 vs. 278.60 ± 230.03, F (2,53) = 7.094, *p* = 0.004, Figure 2d,e). After PPX treatment, the total nocturnal activity in PD monkeys decreased (2752.40 ± 1906.57 vs. 2042.56 ± 1060.97, F (2,53) = 1.289, *p* = 0.440, Figure 2c). Specifically, and the activity 21:00–22:00 and 22:00–23:00 decreased significantly (257.30 ± 284.26 vs. 127.75 ± 169.27, F (2,53) = 3.482, *p* = 0.202; and 278.60 ± 230.03 vs. 102.56 ± 104.06, F (2,53) = 7.094, *p* = 0.01, Figure 2d,e).

### 3.3. PD Monkeys Showed Deficits in AFT Task

The duration of observation in the AFT task was significantly longer in PD monkeys (0.99 ± 1.13) compared to healthy control monkeys (0.530 ± 0.165, F (2,142) = 5.954, *p* = 0.001, Figure 3b). Similarly, the duration of execution was significantly longer in PD monkeys (0.830 (0.200)) than in control monkeys (0.53(0.13), H (2) = 87.75, *p* = 0.000, Figure 3c). After PPX treatment, the durations of observation and execution were significantly reduced (0.67± 0.25, F (2,142) = 5.954, *p* = 0.020, Figure 3b; 0.53 (0.14), H (2) = 87.747, *p* = 0.000, Figure 3c). Note that when food was on one side, the PD monkeys only used the normal limb, no matter which side the food was placed on. The control monkeys used both limbs to obtain the food. The monkeys obtained the food every time.

### 3.4. PPX Reduced Anxiety-Like Behaviors in PD Monkeys

The behavioral responses during the HIT tests are shown in Figure 4. Overall, the anxiety-like behaviors were related to the human intruder’s face orientation. Compared to the control monkeys, the PD monkeys spent less time sitting during the profile, stare, and back phases (profile: 120.00(2.56) vs. 51.93(60.27), *p* = 0.014; stare: 118.87(12.93) vs. 66.90(97.30), *p* = 0.139; and back: 120.00(10.50) vs. 80.00(77.22), *p* = 0.100; H (11) = 21.695, *p* = 0.027; Figure 4e). Instead, they exhibited more locomotor movement ((profile: 0.00(2.57) vs. 15.53(33.09), *p* = 0.153; stare: 1.13(8.98) vs. 0.00(39.29), *p* = 0.342; and back: 0.00(3.67) vs. 36.47(65.10), *p* = 0.009; H (11) = 16.350, *p* = 0.129; Figure 4d). They spent less time in the back of the cage when facing the human intruder (profile: 120.00(51.77) vs. 51.93(61.01), *p* = 0.455; stare: 111.27(38.94) vs. 0.00(64.70), *p* = 0.007; and back: 120.00(114.69) vs. 0.00(0.00), *p* = 0.007; H (11) = 35.411, *p* = 0.000; Figure 4f). After PPX treatment, the total movement, sitting time, or time in the back of the cage in PD monkeys did not change (Figure 4d–f).

An accumulation score of anxiety-like behaviors was used to evaluate the anxiety level in each test phase (see Table 1). The PD monkeys showed higher anxiety-related accumulation scores across testing phases, especially during the profile and stare phases (profile: 0.00(2.00) vs. 5.00(7.00), *p* = 0.006; stare: 2.00(5.00) vs. 6.00(10.00), *p* = 0.246; back: 0.00(1.00) vs. 5.00(5.00), *p* =0.003, H (11) = 25.978, *p* = 0.007; Figure 4g). PPX treatment did not significantly decrease the score (profile: 5.00(7.00) vs. 3.00(3.00), *p* = 0.140; stare: 6.00(10.00) vs. 5.00(5.00), *p* = 0.963; back: 5.00(5.00) vs. 1.00(4.00), *p* = 0.046, H (11) = 25.978, *p* = 0.007; Figure 4g).

In the NFT test, the rates of consuming novel fruit in the control monkeys and PD monkeys were 100% and 80%, respectively. The consumption rate of familiar food was 100% in all monkeys. After PPX treatment, the consumption rates of the PD monkeys increased to 100%. 

In the PCT test, when the predator model (a snake model) was presented, the consumption rates for the control monkeys and PD monkeys were 80% and 40%, respectively. The withdrawal rates were 20% and 60% respectively. 

## 4. Discussion

We conducted a quantitative study to examine non-motor symptoms in monkeys. The MPTP-induced monkeys displayed various degrees of non-motor deficits, such as impairments in working memory, novelty seeking, abnormal sleep, and anxiety-like behaviors. Previously, we reported that these monkeys struggled with learning the DMTS task [16], and their deficits in working memory and sleep patterns were similar to those observed in PD patients [4]. The duration of observation and execution were significantly increased in the MPTP-induced monkeys. In the physical activity monitor test, we found that the nocturnal activity of the MPTP-induced monkeys was higher, especially at 21:00 h and 22:00 h. These results align with previous studies that the MPTP-induced monkeys replicated the fragmented sleep observed in PD [30,31]. To assess anxiety-like behaviors, we used the human intruder test, in which the monkeys were presented with an unfamiliar human intruder [24]. During this test, the monkeys displayed defensive or aggressive behaviors, such as increased movement and reduced time spent sitting, particularly during the profile and stare phases. The accumulation score of anxiety-like behaviors indicates that the PD monkeys were more anxious compared to the control monkeys. Additionally, the MPTP-induced monkeys consumed fewer fruits in the novel food test and significantly less food in the predator confrontation test (40% compared to 80% in control monkeys), showing higher levels of anxiety. Interestingly, the MPTP-induced monkeys also showed a fear of rubber snakes which they usually do not exhibit, and they refused to take food. These findings suggest that mood-related disorders can be induced in MPTP-treated cynomolgus monkeys.

PD is a complex neurodegenerative disease, and the causes of the disease are still not well understood [32]. Non-human primate (NHP) PD models are essential for the discovery and development of drugs to treat Parkinson’s disease. While new animal models have emerged in past years, such as virus-based transgenic techniques or genetically edited monkey models, the MPTP-induced PD model remains the most widely used method to mimic the symptoms of PD. MPTP itself is not toxic, but it is able to cross the blood–brain barrier (BBB) and is quickly converted into a potent dopaminergic neurotoxin called 1-methyl-4-phenylpyridinium ion (MPP+) by the enzyme monoamine oxidase B in astrocytes. MPP+ is selectively taken up by dopaminergic neurons through the dopamine transporter (DAT) due to its structural similarity to dopamine. Subsequently, MPP+ inhibits mitochondrial complex I, leading to the production of reactive oxygen species (ROS) and ultimately causing apoptosis and necrosis in dopamine neurons [6]. The MPTP model can replicate dopaminergic cell loss in the striatum and also reproduce non-motor symptoms of PD, including cognitive, sleep, and gastrointestinal dysfunction [7].

According to previous research, it has been found that chronic neuroinflammation and activated microglia play crucial roles in the neurodegeneration observed in PD [33]. Chronic inflammation can harm the BBB and activate various central nervous system (CNS) cells, such as glial cells, T-cells, and mast cells [34]. Studies have shown that activated microglia can surround damaged neurons and potentially act as antigen-presenting cells, leading to the recruitment of lymphocytes to the brain and triggering an inflammatory reaction [35,36]. This mutual activation of inflammation in both the CNS and peripheral immune cells leads to the release of neurotoxic molecules and exacerbates neurodegeneration [34]. In the MPTP monkey model, it has been observed that dopamine depletion in the putamen is irreversible; even after 5.5–15 years of exposure to MPTP, neuroinflammation cause continuing neurodegeneration [37]. The cognitive deficits caused by MPTP are a result of disruptions in fronto-striatal circuits and are independent of motor dysfunction symptoms [38,39,40]. Interestingly, anxiety-like behaviors have not been observed in rodents treated with MPTP [10,41,42,43]. However, MPTP-induced sleep disorders have been observed, mimicking the sleep disturbances seen in people with PD. The dysregulation of the dopamine metabolism significantly affects the organization of the sleep-wake cycle in the early stages of Parkinsonism [31]. The MPTP-induced model exhibits abnormal psychotomimetic behaviors [44] but not mood-related behaviors [45] or social behaviors [41].

Anti-inflammatory and antioxidant approaches have been the most widely utilized approaches to date [36]. Pramipexole is a type of medication that acts as a presynaptic dopamine agonist. It has a strong affinity for the D3 receptor subtype [46] and also possesses significant intrinsic antioxidant properties [18]. After treatment with Pramipexole, some of the non-motor deficits in the animal model induced by MPTP were partially reversed. However, working memory did not improve, and nighttime activity was reduced to the level seen in the control group, particularly at 21:00 and 22:00 h. This suggests that Pramipexole improved the sleep quality of the monkeys treated with MPTP. Pramipexole also significantly decreased the duration of execution in the AFT task and slightly reduced anxiety accumulation scores in PD monkeys. In a study by M.M et al. (2006), it was shown that Pramipexole can protect nigral dopamine neurons in vivo from the toxicity of MPTP [47]. Pramipexole plays a role in treating both the motor and psychiatric symptoms of PD. There are several arguments supporting the benefits of dopamine agonists over levodopa in early PD treatment. Dopamine agonists are better at slowing down dyskinesias and wearing off. Their advantages result in greater therapeutic benefits in long-term treatment protocols whether used as monotherapy or as an adjunctive therapy to levodopa, while also reducing adverse effects and treatment costs [48]. Furthermore, apart from alleviating motor symptoms, it has been discovered that PPX has anti-anxiety, anti-depression, and paresthesia effects in PD patients [47]. These effects are achieved by targeting the dopamine system and other monoamine pathways, such as the serotonin and the noradrenergic systems.

Our study has certain limitations. First, we used a unilateral MPTP-induced model which may not completely replicate the clinical and pathological aspects of Parkinson’s disease in humans. The systemic MPTP treatment in this model made it challenging for the monkeys to eat, drink, and perform behavioral tasks due to difficulties in swallowing. Secondly, our study focused on an 8-year post-treatment model, which led to the improvement of systemic symptoms in the animals. However, the mood disorder resulting from the physical impairment could not be avoided. Unfortunately, the animals with cognitive impairments faced difficulties in data collection due to the decline in their learning abilities. We made every effort to accurately represent their behavior as much as possible. Moreover, we did not investigate the extent of brain damage in the regions associated with the PD model, nor did we compare the behavioral changes from a pathological perspective. Consequently, it is challenging to quantify the severity of symptoms based on our behavioral representations. Finally, we did not explore the combined effects of pramipexole with other dopamine agonist treatments. Our main focus was to verify whether pramipexole improved non-motor symptoms. Nonetheless, our comprehensive evaluation of non-motor symptoms in MPTP-treated monkeys using various behavioral methods provides valuable insights. Our findings demonstrate that these monkeys exhibit non-motor impairments similar to the symptoms observed in patients with Parkinson’s disease. We firmly believe that this monkey model of PD induced by MPTP will play a significant role in the future in the discovery and development of drugs targeting non-motor symptoms.

## 5. Conclusions

In summary, the aim of this study was to comprehensively evaluate the non-motor symptoms in a PD model using multiple behavioral techniques. The results revealed that MPTP-induced monkeys displayed different levels of non-motor impairments, such as working memory, novelty seeking, abnormal sleep, and anxiety-like behaviors. The administration of pramipexole, a dopamine receptor agonist, was found to alleviate these non-motor symptoms. These findings provide a basis for future behavioral investigations and additional.

## Figures and Tables

**Figure 1 brainsci-13-01153-f001:**
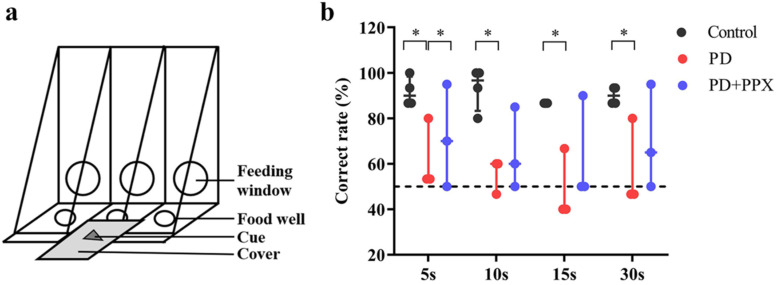
(**a**) Device of delayed matching-to-sample (DMTS) task. (**b**) PD monkeys showed much lower correct rates. PPX treatment did not improve the performance in PD monkeys. *, *p* < 0.05.

**Figure 2 brainsci-13-01153-f002:**
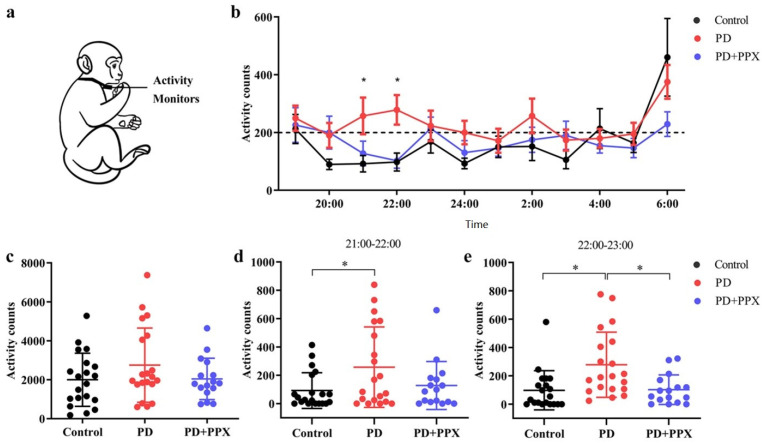
(**a**) Monkeys wearing activity monitors to record activities. (**b**) Hourly activity from 19:00 to 7:00. (**c**) The PD monkeys showed slightly higher total nocturnal activity than the control monkeys, but the difference was not statistically significant. (**d**,**e**) An hour-by-hour analysis showed that the nocturnal activity of the PD monkeys was significantly higher between 21:00 and 23:00. PPX treatment decreased the activity to control level. *, *p* < 0.05.

**Figure 3 brainsci-13-01153-f003:**
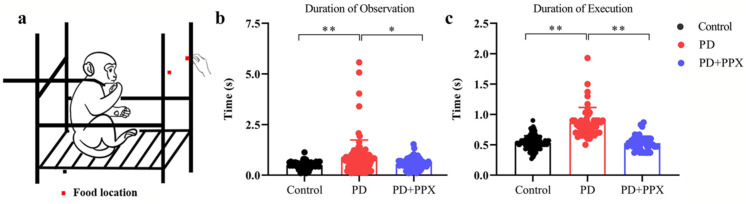
(**a**) Device of the apathy feeding (AFT) task. (**b**) Observation time in AFT task. (**c**) Execution time in AFT task. PD monkeys spend relatively long periods of time in observation (**b**) and execution (**a**). PPX treatment significantly decreased both latencies. *, *p* < 0.05, **, *p* < 0.001.

**Figure 4 brainsci-13-01153-f004:**
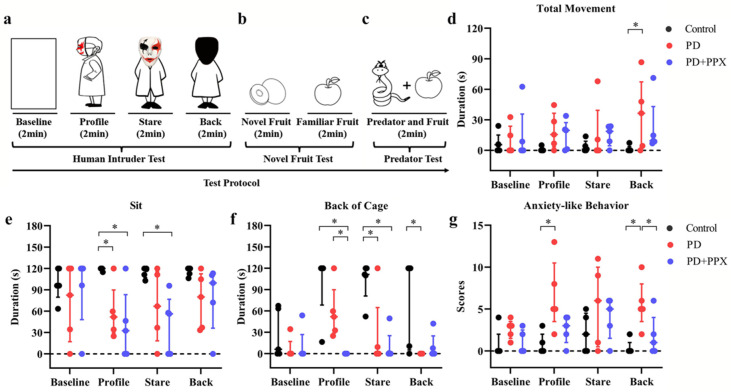
(**a**–**c**) The experimental protocols for testing anxiety, including the human intruder test (HIT) (**a**), novel fruit test (NFT) (**b**), and predator confrontation test (PCT) (**c**). (**d**) Movement duration during tests. PD monkeys moved more during the tests. (**e**) Sitting time during HIT tests. PD monkeys showed less sitting time in the profile and stare phases. (**f**) Time spent in the back of the cages during tests. PD monkeys showed less time staying in the back of the cages. (**g**) Anxiety-like behavior during tests. Accumulation scores showed that PD monkeys were more anxious during the tests. *, *p* < 0.05.

**Table 1 brainsci-13-01153-t001:** Observational behavioral scoring.

Behavior	Evaluation Indicator
Grind teeth	Less than 30 s, for 1 point 30–60 s, for 2 points 60–90 s, for 3 points 90 s or more, for 4 points
Open mouth	Less than 10 s, for 1 point 10–30 s, for 2 points 30–60 s, for 3 points 60 s or more, for 4 points
Tactile/oral exploration, scratch, freeze	Less than 2 s, for 1 point 2–5 s, for 2 points 5–10 s, for 3 points 10 s or more, for 4 points
Yawn, pounce, over one’s eyes, turn around, fear grimace, cage shake	1 time, for 1 point 2 times, for 2 points 3 times, for 3 points 4 times or more, for 4 points

## Data Availability

The data and materials are available upon direct request via the corresponding authors’ emails.

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
