# Peer review of "Quantification of Non-Motor Symptoms in Parkinsonian Cynomolgus Monkeys"

_brainsci, 2023, doi:10.3390/brainsci13081153_

Round 1

Reviewer 1 Report

Comments and Suggestions for Authors

Thank to the authors for their research project.

I have several major concerns:

1- The structure of the introduction is not appropriate. The authors should provide more explanations about Parkinson's disease (first paragraph) and PPX (third paragraph).

2- The aim of the study should be mentioned at the end of the introduction.

3- Please provide a reference for drug doses and treatment duration (section 2.2.).

4- I have problem with the discussion. This section is a repetition of the results section.

5- I would like to read more information about MTPT and its possible mechanisms to induce PD in the discussion.

6- In the discussion, there is no information about PPX and its possible mechanisms. The authors should discuss potential mechanisms underlying PPX effects.

7- How does PPX affect nighttime activity and anxiety? Can authors discuss it?

8- The authors should discuss their results more, not just refer to previous reports.

9- The article has no conclusion. It’s better to add a conclusion.

10- I think the figures should be moved to the results section.

Author Response

Dear Reviews,

Thank you very much for your  professional advice. These comments are all valuable and helpful for improving our article. According to the associate reviewers’ comments, we have made extensive modifications to our manuscript  to make our results convincing. In this revised version, changes to our manuscript were highlighted within the document by using yellow-colored text. Point-by-point responses to reviewers are listed in a Word/PDF file. Please see the attachment

Sincerely.

Yu BAO

Reviewer 2 Report

Comments and Suggestions for Authors

The manuscript titled “Quantification on non-motor symptoms in Parkinsonian cynomolgus monkeys” aims to study non-motor symptoms in MPTP animal model of Parkinson’s disease (PD) in cynomolgus monkeys. The authors have used a wide battery of cognitive and behavioral tests to evaluate non-motor functions in baseline, disease and treatment states. They have used high doses of a dopamine agonist, pramipexole (PPX) as a treatment option. The study showed that the MPTP-treated monkeys exhibited cognitive deficits, abnormal sleep, and anxiety-like behaviors as compared to the control monkeys. These symptoms were relieved partially by PPX. They conclude that MPTP-induced PD monkeys displayed non-motor symptoms that were similar to those found in PD patients. PPX treatment showed moderate therapeutic effects on these non-motor symptoms. 

The study design looks comprehensive and behavioral test battery looks wide and some of the results are robust. The manuscript is written with a clear language with nice figures. However, this reviewer has some major concerns about the animals used for the study, treatment dosage, duration and interpretation of the results:

1-    Animals were lesioned for 8 years ago. This might lead some problems regarding level of neurodegeneration, contribution of the aging, previous treatments have had etc. The authors have to discuss these issues and raise any possible problems/confounders that might cause problems in the results. This reviewer highly suggests at least 2 paragraphs of limitations of the study to the discussion section. 

2-    The dosage of the pramipexole is approximately 10fold compared to humans when calculated as mg per body weight. What was the reason for this? The up-titrating regimen was also significantly different than clinical use. Quick dose increases might cause side effects and lead to under-estimation of the results if animals are not feeling well because of hypotension, nausea etc. Did authors observed any side effects during the treatment phase and had to adjust the dosing?

3-    As a minor point numbering of subtitles for behavioral tests could have been organized under the same subtitles with further sub-numbering for easier read.

4-    This reviewer thinks that the study is lacking motor tests as a confirmation of the lesion and treatment effect. Furthermore, the reason choosing PPX should also be explained. Why not L-dopa which could have given better results instead av partial improvement observed here

Author Response

(The authors gave the same response as above.)

Reviewer 3 Report

Comments and Suggestions for Authors

Authors present an elaboration on non-motor deficits in monkey models treated with MPTP.  The work could be further improved by:

1. Enriching the introduction on Parkinson's disease (PD) pathogenesis. It would be valuable to acknowledge the links between neuroinflammation and neurodegeneration -

Ref.

Platelet-to-lymphocyte ratio and neutrophil-tolymphocyte ratio may reflect differences in PD and MSA-P neuroinflammation patterns. Neurol Neurochir Pol. 2022;56(2):148-155. doi: 10.5603/PJNNS.a2022.0014. Epub 2022 Feb 4. PMID: 35118638.

History of innate immunity in neurodegenerative disorders. Front Pharmacol. 2011 Dec 2;2:77. doi: 10.3389/fphar.2011.00077. PMID: 22144960; PMCID: PMC3228970.

2. Authors refer to interventions using pramipexole. It would be valuable to elaborate on possible feasibility of other dopamine-agonists.

3. The analysis could be presented in a table and more detailed.

4. The limitations of the study should be additionally discussed.

5. The English should be extensively revised. 

Author Response

(The authors gave the same response as above.)

Round 2

Reviewer 1 Report

Comments and Suggestions for Authors

Thanks to the authors for their efforts. My comments have been addressed. The manuscript can be accepted in the present version.

Reviewer 2 Report

Comments and Suggestions for Authors

Thanks for clarifying the points and updating the manuscript

Reviewer 3 Report

Comments and Suggestions for Authors

I do not have further comments.